# Minimizing the Main Strains and Thickness Reduction in the Single Point Incremental Forming Process of Polyamide and High-Density Polyethylene Sheets

**DOI:** 10.3390/ma16041644

**Published:** 2023-02-16

**Authors:** Nicolae Rosca, Mihaela Oleksik, Liviu Rosca, Eugen Avrigean, Tomasz Trzepieciński, Sherwan Mohammed Najm, Valentin Oleksik

**Affiliations:** 1Faculty of Engineering, “Lucian Blaga” University of Sibiu, Victoriei Bd. 10, 550024 Sibiu, Romania; 2Department of Manufacturing Processes and Production Engineering, Faculty of Mechanical Engineering and Aeronautics, Rzeszow University of Technology, al. Powst. Warszawy 8, 35-959 Rzeszów, Poland; 3Kirkuk Technical Institute, Northern Technical University, Kirkuk 36001, Iraq; 4Department of Manufacturing Science and Engineering, Budapest University of Technology and Economics, Műegyetemrkp 3, H-1111 Budapest, Hungary

**Keywords:** incremental forming, main strains, thickness reduction, polyamide, high-density polyethylene, signal-to-noise, analysis of variance

## Abstract

Polymeric materials are increasingly used in the automotive industry, aeronautics, medical device industry, etc. due to their advantage of providing good mechanical strength at low weight. The incremental forming process for polymeric materials is gaining increasing importance because of the advantages it offers: relatively complex parts can be produced at minimum cost without the need for complex and expensive dies. Knowing the main strains and especially the thickness reduction is particularly important as it directly contributes to the mechanical strength of the processed parts, including in operation. For the design of experiments, the Taguchi method was chosen, with an L_18_ orthogonal array obtained by varying the material on two levels (polyamide and polyethylene) and the other three parameters on three levels: punch diameter (6 mm, 8 mm and 10 mm), wall angle (50°, 55° and 60°) and step down (0.5 mm, 0.75 mm and 1 mm). The output parameters were strain in the x direction, strain in the y direction, major strain, minor strain, shear angle and thickness reduction. Two analyses were conducted: signal-to-noise ratio analysis with the smaller-is-better condition and analysis of variance. The optimum values for which the thickness was reduced were the following: wall angle of 50°, punch diameter of 10 mm and step down of 0.75 mm.

## 1. Introduction

The single point incremental forming process (SPIF) emerged in response to the need to manufacture parts in small series and one-off productions without having to use complex and expensive dies. Thus, it was found that by moving a punch along a programmed trajectory, parts of complex configuration could be obtained. A presentation of the process as well as the technological variants of formability in SPIF has been provided by Ben Said [1]. A study on the influence of the process parameters in SPIF is provided by Magdum and Chinnaiyan [2]. Particular importance is also attributed to the sustainability of this unconventional forming process by Liu et al. [3]. In the meantime, a number of technological variants of the incremental forming process have emerged, such as the two-point incremental forming [4], in which the material is supported by a second element on the opposite side to the one on which the punch works, and the use of water as the active element instead of a rigid punch [5]. Naturally, the first pieces of research on this forming process were carried out on metallic materials, in particular deep-drawing steels and aluminum, in the early 1960s. With the desire to obtain parts with satisfactory mechanical strength while keeping a low weight, this process was further applied, in the same way as the conventional material forming processes, to polymeric materials. The first research studies on these materials appeared in the early 2000s. An overview of the main achievements in the field of SPIF processing of polymers can be found in the work of Hassan et al. [6].

Of particular interest with regard to forming processes in general and SPIF in particular is the influence of the process parameters on the main strains and thickness reduction [7,8]. Knowing the strain distribution also allows the evaluation of the failure mode in incremental forming and, implicitly, the determination of formability [9]. The most widely used method for determining strains and thickness reduction in SPIF is the optical, noncontact method, which allows the measurement of these values either continuously, throughout the entire process or only at the end of the forming process [10,11].

Since the present paper deals with strains and thickness reduction for the SPIF of polyamide and polyethylene, only research strictly related to polymeric materials will be referred to in the following. One of the first papers to present the potential and applications of polymeric materials in SPIF is the work of Marques et al. [12]. A more recent study on the behaviour of thermoplastic materials is presented by Zhu et al. [13].

Polymeric materials that can be processed by SPIF are diverse, starting from materials such as polyamide and polyethylene [14], polyvinylchloride [15] and polycarbonate [16] to composite materials having as the basic matrix one of the aforementioned materials reinforced with different fibers (most often glass fiber) [17]. The percentage of glass fiber varies, with the most recent studies analyzing the behaviour of these materials during SPIF when using polyamide 6 reinforced with 30 wt.% glass fiber [18]. All these studies involving composites containing fibers have been carried out under hot conditions, as they cannot be deformed at ambient temperatures. Borić et al. also investigated the formability during SPIF of a composite material made of polyamide with Montmorillonite (MMT) filler clay [19]. A study on the influence of the incremental forming process on the mechanical properties of parts made of amorphous polyvinyl chloride (PVC) and semicrystalline polyamide sheets processed by SPIF is presented by Davarpanah et al. [20]. The authors reported a chain reorientation at the end of the forming process of polymeric materials.

For the SPIF manufacturing of polymer parts as well as other materials, computer numerical control (CNC) milling machines can be used [21]. In the case of the SPIF processing of polymeric materials, where the deformation forces are much lower, the use of industrial robots is also possible [22] despite the main disadvantage of the robot, which is low rigidity compared to CNC milling machines.

Formability in forming processes and particularly in the SPIF process is an area of great interest to researchers. Regarding the formability of polycarbonate sheets in SPIF, a study has been carried out for conical- and pyramidal-shaped parts with varying and fixed slope angles [23]. The authors of this paper analyzed formability with regard to the wall angle at which defects occur in parts processed by SPIF, taking into account three categories of defects: failure, excessive thinning and the twist defect. Rosa-Sainz et al. evaluated the failure modes of polycarbonate sheets by comparing the results obtained from the Nakajima test with those obtained in SPIF [24]. They also presented ways in which failure occurs in SPIF based on the main process parameters of SPIF. A summary concerning the formability of several biocompatible and non-biocompatible materials is presented by Bagudanch et al. [25]. The materials considered were polycarbonate, polypropylene, polyvinylchloride, polycaprolactone and ultrahigh molecular weight polyethylene.

Many of the studies related to the SPIF behaviour of polymeric materials are based on numerical simulation using the finite element method. Some of these studies deal with the SPIF behaviour of these materials at room temperature [26] and are carried out with models that allow the integration of the thermal behaviour of these materials [27].

Due to the advantages of polymeric materials on the one hand and of the SPIF process on the other, SPIF-processed parts have a wide application in a variety of fields. For example, the canopy of various aircraft can be obtained from polycarbonate sheets [28] or the hole flange can be obtained from polyethylene terephthalate [29]. Polymer parts manufactured by SPIF have recently found application in the medical device and medical prosthesis industries [30,31]. For these reasons, Lozano-Sánchez et al. studied the thermal and mechanical behaviour in SPIF of two biocompatible materials: polycaprolactone and ultrahigh molecular weight polyethylene [32]. They considered four process parameters, namely the punch diameter, rotational speed, feed rate and step down, and used a Box-Behnken design of experiments. The thermal properties of the two biocompatible materials were studied using the Vicat test. The results of the research were centered on the forces in the process and on the temperature released as a result of the processing of the parts.

Other studies have focused on identifying the types of defects that occur in polymer parts processed by SPIF. Thus, Al-Ghamdi studied one of the most important occurring defects contributing to shape and dimensional deviations of SPIF-processed parts, that is, the springback [33]. He studied the influence of the wall angle, feed rate, rotational speed, punch diameter and step down on the springback of parts made of polypropylene sheets. Another paper studied the wrinkle defect in polytehylene parts [34]. The geometrical accuracy of polymer parts made by SPIF was analyzed by Maaß et al. [35]. They recommended increasing the initial thickness of the part and increasing the punch radius for higher accuracy of parts made of thermoplastic materials.

The influence of the process parameters, whether they are parameters related to the processed part or parameters related to the forming technology, has also been studied. For instance, the influence of the input parameters on the surface roughness, springback and thinning has been analyzed for polypropylene [36] and polyvinylchloride sheets [37]. Kharche and Barve used the Taguchi method to investigate the influence of the rotational speed, feed rate, step down and wall angle on the accuracy of parts manufactured by SPIF and made of polyamide 6, with direct reference to the final thickness, final depth of the part and wall angle deviation [38]. The influence of the feed rate, rotational speed, punch diameter and step down direction on the maximum force, surface roughness and final depth of the part for two biocompatible materials was analyzed by Sabater et al. [39]. Thangavel et al. studied the influence of the punch diameter on the formability and thinning of four polymeric materials. The authors also analyzed the microstructure of SPIF-processed parts and the influence of the punch diameter on the respective microstructure [40].

Among the most important output parameters of SPIF are the forces and deformations occurring during the forming process. The study of the variation of forces in the SPIF manufacturing of polymeric materials has been carried out either by simulation using the finite element method for glass polymers and polyvinylchloride and polycarbonate sheets [41] or by experiment for polyamide and polyethylene [14]. The work of Rai et al. allows the prediction of forces and the calculation of the fracture energy based on a numerical simulation for the frustum of pyramid-shaped parts made of polycarbonate [42].

Wei et al. correlated the plastic strain induced in polypropylene parts processed by SPIF to the different post-forming tensile properties [43]. The elasticity modulus, elongation, yield stress, ultimate stress and drawing stress were analyzed.

Harhash and Palkowski’s paper uses digital image correlation to study the main strains and thickness reduction that occur during the SPIF process in steel/polymer/steel sandwich composite sheets [44]. They used a thermoplastic polyolefin polymeric material of two thicknesses and two galvanized steels. The parts made were of the frustum of a cone type, with variable wall angle. Based on the measurement of main strains and thickness reduction, the formability of the sandwich materials was evaluated.

In reviewing studies related to the evaluation of the main strains and thickness reduction of polymeric materials in the SPIF process, it became apparent that there are no comprehensive studies that take into account the influence of as many parameters as possible related to the part geometry or the processing technology of polymeric materials. In the present paper, the aim was to study the influence of the most important parameters, namely the punch diameter, step down, wall angle and material type on the most important strains that occur as a result of the SPIF process: strain in the x and y directions, major strain, minor strain, shear angle and thickness reduction. In contrast to other studies in the literature, the variation of strains was determined throughout the entire forming process, not just at the end of the process. Two materials (polyamide and high-density polyethylene) have been chosen for their low density, relatively good mechanical properties and low cost compared to other polymeric or composite materials, making them suitable for a wide range of applications, such as aircraft parts, automotive parts, parts used in the medical equipment industry, etc. For the other input parameters, three levels of variation were chosen. It was our intention not only to highlight the influence of each input parameter on the output parameters but also to provide a guideline for minimizing the main strains of parts manufactured by SPIF made of polyamide and high-density polyethylene. For these purposes, by means of the Taguchi method for the design of experiments and analyses including signal-to-noise analysis and analysis of variance, the optimal values of the input parameters were identified in order to achieve the lowest possible thickness reduction. For the two chosen materials, linear regression relationships for the thickness reduction were also developed.

## 2. Materials and Methods

To carry out the experimental research of the present work, it was necessary to choose a machine that would perform single point incremental forming of polyamide and polyethylene materials. The options included the use of a CNC milling machine or an industrial robot. The CNC milling machine allows higher forces under increased stiffness conditions than the industrial robot but does not allow the visualization and, therefore, measurement of the main strains and thickness reduction throughout the manufacturing process. The aim of the paper is the measurement of the main strains and thickness reduction during the entire forming process, which was not possible on a CNC milling machine, as the optical cameras of the measuring system could not be positioned under the milling machine table. Moreover, in the case of polymeric materials, the forces that occur in the process are much smaller than those occurring in the SPIF processing of metallic materials. Given the advantage of being able to measure the main strains and thickness reduction without affecting the rigidity of the machine, the Kuka KR210-2 robot was chosen. Choosing this robot allowed the use of a vertical blanksheet mounting bracket as well as the positioning of the optical measuring system cameras in such a way that the area that does not come into contact with the blanksheet is visible throughout the manufacturing process (Figure 1).

The Kuka KR210-2 robot is a 6-axis robot that can develop a maximum load of 2100 N and is equipped with a KR C2 controller. In order to be able to generate the desired trajectory, it was necessary to create a three-dimensional model of the part to be manufactured and then export the stp standard of the model to the SprutCam program. Two types of trajectories were developed: a frustum of a cone (Figure 2a), used to process all the parts in the Taguchi design of experiments, and a frustum of a pyramid (Figure 2b), used for comparison of two cases in the Taguchi design of experiments. Both types of trajectories were of the spiral type, i.e., with continuous feed in the vertical direction and no indentation zones, as it was found from previous studies that increased local deformations occur in these zones.

The optical system, Aramis, is a noncontact measuring system that allows the measurement of strains and thickness reduction throughout the entire forming process [45]. This is achieved by applying a light-coloured, matt paint to the surface that does not come into contact with the punch, followed by splashes of another dark-coloured, matt paint (for contrast) after the first layer of paint has dried. The two paints must be matt in order to avoid reflecting light during the measuring process.

The Aramis optical system included two Zeiss optical cameras with a focal length of 50 mm and an aperture adjustment range from f2.8 to f11, achieving a maximum image resolution of 1600 × 1200. The software with which the optical system is equipped (Aramis) measured the displacement of each speck of paint on the surface of the part and then converted these displacements into strains as well as into thickness reductions based on the law of constant volume. The strain in the x direction, strain in the y direction, major strain, minor strain, shear angle and thickness reduction were measured.

Hemispherical punches with three sizes of the active zone (6 mm, 8 mm and 10 mm) were used for the experimental research. These were heat-treated and preground to a roughness of Ra = 0.8 μm. The system fastening the punches to the robotic arm featured an elastic bushing, enabling easy replacement of the punches.

The blanksheets were positioned vertically and were clamped between two plates during the single point incremental forming process by means of 12 hex head screws. The blanksheets made of polyamide 6.6 (PA) and high-density polyethylene 1000 (HDPE) were all 3 mm thick, square in shape and with a side length of 250 mm. An area of 200 × 200 mm^2^ of the fastening system remained visible, which was more than sufficient since the lower base of the frustum of the cone was 85 mm.

The main differences between the different thermoplastic materials lie in the different arrangements of the molecular chains. These different arrangements lead to both different thermal exposure behaviour of these materials and different mechanical properties. Thus, the two polymeric materials used in the present paper are distinct in terms of structure. HDPE has a semicrystalline structure, while PA 6.6 has a crystalline structure, which provides it with good electrical resistance as well as good resistance to high temperatures. A characteristic of PA 6.6 is that it is made of two monomers, namely adipoyl chloride and hexamethylene diamine, which give it a crystalline structure. Given its crystalline structure, PA has a better surface quality and better processability than HDPE due to its lower viscosity. The polymer chains in HDPE are considered to be more linear. They are found closer together, resulting in higher intermolecular forces and hence higher mechanical strength values.

Both materials, PA and HDPE, are materials with high elongation at break and, in order to highlight their mechanical characteristics, they were subjected to uniaxial tensile testing in a previous paper [14]. Thus, for PA, the following mean values were obtained: for Young’s modulus, E = 1827.3 MPa; for maximum tensile stress, σ_max_ = 41.5 MPa; and for tensile strain at maximum tensile stress, ε_max_,_Ts_ = 126.5%. The following mean values were obtained for HDPE: for Young’s modulus, E = 1004.8 MPa; for yield stress, Y_s_ = 12.6 MPa; for maximum tensile stress, σ_max_ = 24.7 MPa; and for tensile strain at maximum tensile stress, ε_max_,_Ts_ = 496.7%. As can be seen from the mechanical characteristics of the polymeric materials, both have high tensile strains, and PA has a higher maximum tensile stress than HDPE as well as a higher value for Young’s modulus. This means that HDPE has a stronger springback than PA.

## 3. Results and Discussions

### 3.1. Main Strains and Thickness Reduction Determination during Incremental Forming of PA and HDPE Sheets for Frustum of Cone-Shaped Parts

A research study on the forming behaviour of certain materials would be incomplete without a comprehensive study on the variation of the main strains and thickness reduction during the forming process. For these reasons, the use of the Kuka KR210-2 industrial robot as a working machine allowed the measurement of deformations during the process, from the first contact of the punch with the material to its withdrawal from the part.

As previously mentioned, Gom’s Aramis optical measuring system was used to measure deformations. This system, in contrast to the Argus system (also from Gom), makes it possible to measure the strains and thickness reduction throughout the entire measurement process, as opposed to only at the end of the process. This system involves measuring the position of small-sized points throughout the measurement process. The software package of the Aramis system then converts the displacement of those points into strains and, through calculations, into thickness reduction.

For the experimental investigations related to the determination of the main strains and thickness reduction during the single point incremental forming of PA and HDPE, the most important influencing factors are presented in Table 1.

On the basis of the bibliographic research presented above and also taking into account the practical realities of carrying out the experimental research, two factors related to the part (polymeric material and wall angle) and two factors related to the technological parameters (punch diameter and step down) were selected.

Due to the number of factors selected (four) and the levels of variation desired (two levels for one factor and three levels for the other three), the Taguchi method was chosen for the design of experiments with the aim of limiting the total number of experiments while identifying designs that are unaffected by changes in the control factors. One of the key principles of the Taguchi design of experiments is the use of signal-to-noise (S/N) ratios to compare the performance of different combinations of control factors. S/N ratios measure the ratio of signal, or desired output, to noise, or unwanted variation. Higher S/N ratios indicate better performance.

Depending on the purpose of the experiments, different signal-to-noise ratios (S/N) can be selected. Three different characteristic S/N formulations (conditions) are present in the following equations.

Smaller is better:(1)$$S/N=-10\cdot log\left[\frac{1}{n}\sum_{i=1}^{n}\left(y_{i}^{2}\right)\right]$$

Larger is better:(2)$$S/N=-10\cdot log\left[\frac{1}{n}\sum_{i=1}^{n}\left(\frac{1}{y_{i}^{2}}\right)\right]$$

Nominal is better:(3)$$S/N=-10\cdot log\left[\frac{\hat{y}}{s^{2}y}\right]$$
where $y_{i}$ is the measured data, $\hat{y}$ is their arithmetic mean, $s^{2}y$ is the standard deviation of these data and n is the number of responses in the factor level combination.

Since it is important in SPIF that the size of the main strains and thickness reduction, which subsequently influences the operational mechanical strength of the part, is as small as possible during the forming process, the “smaller is better” option was chosen.

Table 2 contains the design of experiments with the factors and levels of variation as well as their codifications. The eighteen experiments were repeated twice in order to reduce the error in the measurement of the main strains and thickness reduction.

As previously mentioned, two factors related to the manufactured part, namely the part material (MAT) and the wall angle (α), and two factors related to the technology, namely the punch diameter (Dp) and the step down (s), were used. For the part material, as mentioned above, PA and HDPE were selected, meaning there were two levels of variation, while the wall angle was selected with three levels of variation (50°, 55° and 60°), the punch diameter was also selected with three levels of variation (6 mm, 8 mm and 10 mm) and the step down was also varied on three levels (0.5 mm, 0.75 mm and 1 mm).

After the design of experiments, experiments concerning the measurement of main strains and thickness reduction were carried out in random order, with each measurement being repeated twice for error reduction.

The numerical results of the two measurements, the mean value of the measurements, the standard deviation, the S/N ratio and the coefficient of variation for the strain in the x direction, ε_x_; strain in the y direction, ε_y_; major strain, ε_1_; minor strain, ε_2_; and shear angle, γ_xy_, are presented in Appendix B, and the thickness reduction, t_r_, is presented in Table 3.

The sequence of Figure 3, Figure 4, Figure 5, Figure 6, Figure 7 and Figure 8 shows the results obtained for the Cases marked S2 (for PA) and S11 (for HDPE) for the same geometric shape of the part (α = 50°) and for the same technological regime (Dp = 8 mm and s = 0.75 mm) for the first measurement given in Table 3 as well as Table A1, Table A2, Table A3, Table A4 and Table A5 (Appendix B).

The analysis of the results obtained from the measurement of the main strains and thickness reduction in SPIF led to the identification of certain patterns.

Thus, it can be said that the strain distributions in the Ox and Oy directions (ε_x_ and ε_y_, respectively) are somewhat similar, of course with a 90° angular index to each other, for both polyamide and polyethylene parts. Their maximum values are located on the conical wall of the part, in the form of a petal, symmetrical to the plane perpendicular to the direction that gives them their name. The local maximum values are obtained near the axis indicating the strain in question, towards the upper base of the frustum of the cone.

The distribution of the major strain ε_1_ is relatively uniform on the cone wall of the deformed part, but the maximum values occur in the end area of the working stroke of the punch, i.e., towards the upper base area of the frustum of the cone, in both polyamide and polyethylene parts. The orientation of the maxima towards the upper base area of the frustum of the cone is more visible the smaller the angle of the part (in parts with a wall angle of 50°).

The minor strain ε_2_ gives maximum values towards the upper base of the frustum of a cone, but in this case, there are differences between polyamide and polyethylene parts in terms of distribution. In polyamide parts, which have higher stiffness, the maximum values of the minor strain are distributed in the form of a ring around the upper base of the frustum of the cone, which at small angles is concentrated right towards the upper base of the frustum. In polyethylene parts, which are less rigid, a local maximum occurs in the area where the punch loses contact with the part at the end of the stroke. For this reason, the distribution of the minor strain in polyethylene parts is not uniform on the conical wall of the part.

The distribution of the shear angle is symmetrical with respect to the xOz and yOz planes, with positive and negative maxima on either side of these two planes for both types of materials analyzed.

The distribution of the thickness reduction is similar to that of the major strain ε_1_ for both types of materials, that is, a relatively uniform distribution on the cone wall with maximum values towards the upper base of the frustum of the cone. However, in polyethylene parts, the presence of local maxima of the minor strain also leads to maxima of the thickness reduction but of lower intensity.

The maximum values of the ε_x_ and ε_y_ strains are always higher for polyamide than for polyethylene when using the same punch, the same step down and parts with the same vertical wall angle. The maximum value of the ε_x_ strain occurs in Case S7 for polyamide (0.8394 mm/mm) and in Case S17 for polyethylene (0.6870 mm/mm). The maximum value of the ε_y_ deformation occurs in Case S7 for polyamide (0.8424 mm/mm) and again in Case S17 for polyethylene (0.7053 mm/mm).

The minimum value of the ε_x_ strains is found in Case S3 for polyamide (0.4618 mm/mm) and in Case S12 for polyethylene (0.4165 mm/mm). The minimum value of the ε_y_ strain is again found in Case S3 for polyamide (0.4617 mm/mm) and in Case S11 for polyethylene (0.4165 mm/mm).

The maximum values of the major strain ε_1_ are also always higher for polyamide than for polyethylene when using the same punch, the same step down and parts with the same vertical wall angle. The maximum value of the major strain ε_1_ is recorded in Case S7 for polyamide (0.8500 mm/mm) and in Case S17 for polyethylene (0.6895 mm/mm). The minimum value of the major strain ε_1_ is found in Case S3 for polyamide (0.4652 mm/mm) and in Case S12 for polyethylene (0.4278 mm/mm).

The maximum values of the minor strain ε_2_ are always higher for polyethylene than for polyamide, contrary to the other types of strains, for cases where the same punch, the same step down and parts with the same vertical wall angle are used. The maximum value of the minor strain ε_2_ is found in Case S3 for polyamide (0.2098 mm/mm) and in Case S12 for polyethylene (0.2336 mm/mm). The minimum value of the minor strain ε_2_ occurs in Case S4 for polyamide (0.0755 mm/mm) and in Case S16 for polyethylene (0.0902 mm/mm).

The maximum values of the thickness reduction t_r_ are mostly, but not always, higher for polyamide than for polyethylene when using the same punch, the same step down and parts with the same vertical wall angle. The maximum value of the thickness reduction t_r_ is found in Case S8 for polyamide (0.9085 mm/mm) and in Case S17 for polyethylene (0.8465 mm/mm). The minimum value of the thickness reduction t_r_ is recorded in Case S2 for polyamide (0.5831 mm/mm) and in Case S11 for polyethylene (0.5827 mm/mm).

The maximum values of the shear angle γ_xy_ are also always higher for polyamide than for polyethylene for cases performed under the same processing conditions. The maximum value of the shear angle γ_xy_ is found in Case S7 for polyamide (40°) and in Case S16 for polyethylene (31.925°). The minimum value of the shear angle γ_xy_ is recorded in Case S3 for polyamide (22.1°) and in Case S12 for polyethylene (15.8°).

Both materials used in the experimental research are, as mentioned above, polymeric, low-density materials. The formability of these types of materials was initially studied on the basis of the strength-differential effect, which is based on the ratio of compressive flow stress and tensile flow stress of the polymeric material, and was later included in the evaluation of formability and pressure-sensitive yield surface. Because of the different structures of the two polymeric materials, as previously stated, their processability is also different. The friction between the hemispherical punch and the polymeric material is lower in the case of PA than in the case of HDPE due to the crystalline structure leading to better surface quality. This also leads to lower deformation forces in the case of PA than in the case of HDPE [14]. Moreover, since HDPE, with its tighter bonds between the molecular chains, is stiffer, the maximum values of the strains in both x and y directions as well as of the major strain, thickness reduction and shear angle are lower than those obtained under the same manufacturing conditions for polyamide. In the case of the minor strain, the values obtained for HDPE are lower than those obtained for polyamide under the same manufacturing conditions. As for the thickness reduction, it is well known that it increases with increasing plastic strain [15], whereas for the frustum of cone-type parts, it has the same variation as that of the major strain.

### 3.2. Analysis of the Signal-to-Noise Ratio for the Main Strains and Thickness Reduction during the Incremental Forming of PA and HDPE Sheets

For all types of strains and for the thickness reduction of the polymeric material, the aim is to minimize the maximum values; therefore, the “smaller is better” condition for the signal-to-noise ratio (S/N) is used. The mean response of the signal-to-noise ratios (S/N) for the ε_x_ strain (with the “smaller is better” condition) is shown in Table 4.

Table 4 shows the mean response values of the S/N ratios, which analyze the effect of the influencing factors (MAT, α, Dp and s) on the strain in the x direction (ε_x_). It shows the optimal levels (based on the S/N ratios) for which the controlling factors result in the lowest value of ε_x_. The following optimal values are found for the part material at level 2 (S/N = 5.492), for the wall angle at level 1 (S/N = 6.587), for the punch diameter at level 3 (S/N = 5.263) and for the step down at level 2 (S/N = 4.803). In terms of rank, for strain ε_x_, the wall angle has the highest impact (delta: S/N = 3.576 and rank = 1), followed by the part material (delta: S/N = 1.577 and rank = 2), punch diameter (delta: S/N = 1.083 and rank = 3) and step down, which has the lowest impact (delta: S/N = 0.227 and rank = 4).

The mean response of the signal-to-noise ratios (S/N) for the strain in the y direction (ε_y_) (with the “smaller is better” condition) is shown in Table 5.

For the strain in the y direction (ε_y_) the optimal values are obtained as follows for the part material at level 2 (S/N = 5.508), for the wall angle at level 1 (S/N = 6.582), for the punch diameter at level 3 (S/N = 5.270) and for the step down at level 2 (S/N = 4.797). In the case of strain ε_y_, as in the case of strain ε_x_, the wall angle has the highest impact (delta: S/N = 3.606 and rank = 1), followed by the part material (delta: S/N = 1.608 and rank = 2), punch diameter (delta: S/N = 1.107 and rank = 3) and step down, which has the lowest impact (delta: S/N = 0.245 and rank = 4). The close values between the two strains ε_x_ and ε_y_ also serve as a “control” of the measurements, with it being absolutely normal, given the geometry of the part, that the values of the two strains and their variation mode are very similar.

The mean response of the signal-to-noise ratios (S/N) for the major strain ε_1_ (with the “smaller is better” condition) is shown in Table 6.

The analysis of Table 6 shows that the optimum values that give the minimum major strain ε_1_ are found for the part material at level 2 (S/N = 5.132), for the wall angle at level 1 (S/N = 6.293), for the punch diameter at level 3 (S/N = 5.127) and for the step down at level 2 (S/N = 4.533). Similar to the ε_x_ and ε_y_ strains, the wall angle also has the highest impact for the major strain ε_1_ (delta: S/N = 3.460 and rank = 1), followed by the part material (delta: S/N = 1.373 and rank = 2), the punch diameter (delta: S/N = 1.249 and rank = 3) and the step down, with the lowest impact (delta: S/N = 0.190 and rank = 4).

The mean response of the signal-to-noise ratios (S/N) for the minor strain ε_2_ (with the “smaller is better” condition) is shown in Table 7.

From the very start of the analysis, it should be noted that the values of the minor strain ε_2_ are much smaller and much closer to each other. Because of this, different optimal levels, different ranks and a different order of magnitude of the signal-to-noise ratio will be observed for the minor strain ε_2_ as compared to the strains analyzed so far. Based on the analysis of Table 7, the optimal values that give the lowest minor strain ε_2_ are found for the part material at level 1 (S/N = 18.88), for the wall angle at level 3 (S/N = 19.95), for the punch diameter at level 1 (S/N = 19.38) and for the step down at level 2 (S/N = 19.23). In the case of the minor strain ε_2_, the wall angle has the highest impact (delta: S/N = 5.10 and rank = 1), followed by the step down (delta: S/N = 2.25 and rank = 2), the punch diameter (delta: S/N = 2.23 and rank = 3) and the part material, which has the lowest impact for this strain (delta: S/N = 1.92 and rank = 4). It can be concluded that in this case, the wall angle is of the greatest importance, whereas among the other three factors, the delta values are very close, making it impossible to draw a definite conclusion on their importance.

The mean response of the signal-to-noise ratios (S/N) for the shear angle γ_xy_ (with the “smaller is better” condition) is shown in Table 8.

Table 8 shows that the optimal values that give the minimum shear angle γ_xy_ are found for the part material at level 2 (S/N = −27.39), for the wall angle at level 1 (S/N = −26.22), for the punch diameter at level 3 (S/N = −27.89) and for the step down at level 3 (S/N = −28.50). In the case of the shear angle γ_xy_, as for the strains in the x and y directions, ε_x_ and ε_y_, and for the major strain ε_1_, the wall angle has the highest impact (delta: S/N = 4.25 and rank = 1), followed by the part material (delta: S/N = 2.34 and rank = 2), punch diameter (delta: S/N = 1.32 and rank = 3) and step down, with the lowest impact (delta: S/N = 0.130 and rank = 4).

The mean response of the signal-to-noise ratios (S/N) for the thickness reduction *t_r_* (with the “smaller is better” condition) is shown in Table 9.

Analysis of Table 9 shows that the optimum values resulting in the lowest thickness reduction, *t_r_*, are found for the part material at level 2 (S/N = 3.146), for the wall angle at level 1 (S/N = 4.141), for the punch diameter at level 3 (S/N = 3.201) and for the step down at level 2 (S/N = 3.283). As with the other strains, the wall angle also has the highest impact on the thickness reduction (delta: S/N = 2.412 and rank = 1), which is almost four times higher than that of the step down (delta: S/N = 0.580 and rank = 2), almost six times higher than that of the part material (delta: S/N = 0.431 and rank = 3) and again almost six times higher than that of the punch diameter, which has the lowest impact (delta: S/N = 0.422 and rank = 4).

Graphical representations (main effects plot for the signal-to-noise (S/N) ratios) of the factor levels in Table 4, Table 5, Table 6, Table 7, Table 8 and Table 9 are shown in Figure 9, Figure 10, Figure 11, Figure 12, Figure 13 and Figure 14, respectively.

An analysis of the graphs in Figure 9, Figure 10, Figure 11, Figure 12, Figure 13 and Figure 14 shows that for all strains, the wall angle of the part has the most important influence because it also has the steepest slope in the main effects plot. With two exceptions, namely the thickness reduction and the minor strain, the second most important factor for the strains obtained in polymer parts processed by incremental forming is the part material. The main purpose of the experimental research is to find those levels of the influence factor that would lead to the minimization of the noise factors on the responses. The optimal parameters leading to reduced strains are easily determined from both Figure 9, Figure 10, Figure 11, Figure 12, Figure 13 and Figure 14 and Table 4, Table 5, Table 6, Table 7, Table 8 and Table 9.

As in the case of forces in incremental forming, the best level for each control factor was found based on the highest signal-to-noise ratio of each influencing factor. According to the analysis of the values in Table 4, Table 5, Table 6, Table 7, Table 8 and Table 9, the optimal values for which the strains ε_x_, ε_y_, ε_1_, t_r_ and γ_xy_ are as small as possible are obtained for the following conditions: α = 50°, MAT—polyethylene, D_p_ = 10 mm and s = 0.75 mm (except for the shear angle for which the optimal value of the step down is s = 1). Given the extremely close delta values for the step down in the case of the shear angle and the fact that the step down has the lowest influence, this does not constitute an issue. The optimum values at which the minor strain, ε_2_, is minimal are as follows: α = 60°, MAT—polyamide, Dp = 6 mm and s = 0.75 mm. As previously mentioned, the aim is that the thickness reduction has minimum values, as the minor strain, which has the lowest values anyway, has a limited influence on the forming process. For this reason, it can be concluded that the optimum values are those obtained for the strains ε_x_, ε_y_, ε_1_, t_r_ and γ_xy_.

It can be seen in the graphs presented in Figure 9, Figure 10, Figure 11, Figure 12, Figure 13 and Figure 14 that the results in Figure 12 show a different trend compared to the others presented in the paper. It should be mentioned ahead that the major strain is obtained in the circumferential direction of the frustum of a cone-type part, while the minor strain is obtained in the meridional direction. Furthermore, plane strain stretching is known to exist on the conical wall, as can be also seen in Figure 5a,b and Figure 6a,b. Plane strain stretching causes the influences of the analyzed parameters to lead to different results. Only the step down has the same influence, as it, in fact, has the least influence on both the major and the minor strains.

Regarding the influences of the parameters considered in relation to the main strains, it can be said that they are in agreement with other studies presented previously. An increase in the wall angle, which has the most influence on the geometric and technological parameters, leads to increased main strains and thickness reduction [12]. As for the step down, increasing its value can lead to greater void densities and changes in the molecular chain. Decreasing its value leads to multiple punch passes and increased thinning [13]. In addition, increasing the punch diameter leads to an increase in the contact area between the punch and the processed part and thus to a decrease in the thickness reduction.

### 3.3. Analysis of Variance (ANOVA) for the Main Strains and Thickness Reduction during the Incremental Forming of PA and HDPE Sheets

Following the analysis based on the S/N ratios, the analysis of variance (ANOVA) was used to determine the contribution of each factor involved. For the assessment of the interactions, those between the material and the technological factors, namely the punch diameter and the step down, were chosen. ANOVA tables were calculated with a 5% significance level and a 95% confidence level. Table 10 shows the results of the ANOVA for the strain in the x direction, ε_x_.

When analyzing the *p*-values in Table 10, it can be seen that a *p*-value greater than 0.05 occurs for the step down and for the two interactions (material × punch diameter and material × step down). Therefore, these factors are not significant for the strain in the x direction, ε_x_.

Table 11 shows the results of the ANOVA for the strain in the y direction, ε_y_.

An analysis of the *p*-values in Table 11 shows that, similar to the strain in the x direction ε_x_, a *p*-value greater than 0.05 also occurs for the step down and the two interactions (material × punch diameter and material × step down). Therefore, all other factors are significant for the ε_y_ strain.

Table 12 shows the results of the ANOVA for the major strain, ε_1_.

As in the case of the Taguchi analysis, the most important factors influencing the major strain ε_1_ are identical to those significantly influencing the strains in the x and y directions, ε_x_ and ε_y_. Based on the *p*-values presented in Table 12, it can be seen that a *p*-value greater than 0.05 is found for the strains in the x and y directions, ε_x_ and ε_y_, the step down and two interactions (material × punch diameter and material × step down).

Table 13 shows the results of the ANOVA for the minor strain, ε_2._

Unlike the other types of strains, in the case of the minor strain ε_2_, all the considered influencing factors are significant because their *p*-values are less than 0.05. The only factors without influence are the two interactions (material × punch diameter and material × step down).

Table 14 shows the results of the ANOVA for the shear angle, γ_xy_.

An analysis of the *p*-values in Table 14 shows that a *p*-value greater than 0.05 occurs in the case of the step down and the two interactions (material × punch diameter and material × step down). Therefore, these factors are not significant for the shear angle γ_xy_.

Table 15 shows the results of the ANOVA for the thickness reduction, *t_r_*.

In contrast to the other types of strains, the analysis of the p-values in Table 15 shows that in the case of the thickness reduction, only one influencing factor is significant, that is, the wall angle of the part. All other influencing factors have values greater than 0.05 (part material—0.166, punch diameter—0.427 and step down—0.257). The two interactions (as expected, since the factors that form them are not significant) also have values above 0.05 (material × punch diameter—0.216 and material × step down—0.296).

The percent contribution in the ANOVA tables shows the magnitude of the influence of the parameters on the main types of strains. The contribution of the wall angle is 70.75%, of the part material is 20.47%, of the punch diameter is 6.46% and of the step down is only 0.3% for the strain in the x direction, ε_x_. The interactions of the material with the two technological parameters, in turn, have an extremely low contribution for the strain in the x direction: 0.26% for the material with the punch diameter and 0.24% for the material with the step down. 

As in the case of the strain in the x direction, for the strain in the y direction (ε_y_), the contributions are 69.75% for the wall angle, 20.69% for the material, 6.55% for the punch diameter and 0.37% for the step down. The interactions of the material with the two technological parameters have, in this case too, extremely low contributions for the strain in the y direction: 0.32% for the material with the punch diameter and 0.26% for the material with the step down. It can be seen that, as expected, the contribution values for the two strains are very close due to the geometry of the part.

In the case of the major strain ε_1_, the contributions are 70.88% for the wall angle, 16.51% for the material, 9.34% for the punch diameter and 0.22% for the step down. The contributions of the interactions are 1.11% for the material with the punch diameter and 0.69% for the material with the step down.

In the case of the minor strain ε_2_, the contributions are 58.93% for the wall angle, 12.85% for the punch diameter, 11.08% for the material and 10.98% for the step down. It can be noticed that apart from the wall angle, the other influencing factors have contributions that are close in value. The contributions of the interactions are also reduced in this case: 0.94% for the part material with the punch diameter and 1.55% for the part material with the step down.

In the case of the shear angle γ_xy_, the contributions are 64.55% for the wall angle, 28.39% for the material, 6.07% for the punch diameter and 0.06% for the step down. The contributions of the interactions are 0.62% for the material with the punch diameter and 0.18% for the material with the step down.

In the case of the thickness reduction, tr, the contributions are 71.38% for the wall angle, 4.7% for the step down, 3.41% for the material and 2.7% for the punch diameter and. The contributions of the interactions are 5.48% for the material with the punch diameter and 4.11% for the material with the step down. It can be seen that in the case of the thickness reduction, the contribution of the wall angle is much higher than that of the other factors. In addition, the contributions of the interactions are larger than the singular contributions of the material and the punch diameter.

It can be concluded that the most important parameter for all types of strains is the wall angle of the part.

### 3.4. Regression Analysis for the Thickness Reduction during the Incremental Forming of PA and HDPE Sheets

A regression analysis was also performed in order to obtain the mathematical dependence relations between the thickness reduction and the input parameters analyzed in the paper (α, Dp and s). The thickness reduction was chosen because, among all strains, it is the most important parameter in terms of the subsequent functioning of the part manufactured by SPIF. The analyses were separated, obtaining the linear regression equations for each type of material. The R-sq (pred) parameters were calculated, which indicate how well the model predicts responses for new measurements, to evaluate the regression model. Following the regression analysis performed with Minitab, the data shown in Table 16 for the thickness reduction, t_r_, for polyamide were obtained.

As can be seen from the values obtained for p, the corresponding coefficients for the step down and punch diameter are insignificant, with values greater than 0.05. The following regression equation is thus obtained:(4)$$tr_{\_PA}=-0.696+0.02610\cdot \alpha$$

The values obtained in the table indicate good predictability of the proposed mathematical model (of 90.47%).

The regression analysis was then performed for the thickness reduction, t_r_, for polyethylene, the results of which are shown in Table 17.

In this case, it is observed that, except for the part angle, all other coefficients are insignificant, with values greater than 0.05. The following regression equation is obtained:(5)$$tr_{\_PE}=0.012740\cdot \alpha$$

In this case, excellent predictability of the proposed mathematical model is observed (of 99.21%).

After studying the signal-to-noise analysis, ANOVA and regression analysis for the thickness reduction, several conclusions can also be drawn: for all types of strains, including the shear angle and the thickness reduction, the greatest influence is given by the wall angle of the processed part. For the ε_x_ and ε_y_ strains and for the major strain ε_1_, and also for the shear angle γ_xy_, the second most important is the material of the part, followed by the punch diameter and the step down. For the minor strain ε_2,_ the second most important factor after the wall angle of the part is the punch diameter, followed by the part material and the step down. For the thickness reduction, the second most important influence is also that of the step down, the same as in the case of the minor strain ε_2_, but the influence of the part material and of the punch diameter is reversed. The ANOVA results show that the step down has a negligible contribution for the strains in Ox and Oy directions, the major strain ε_1_ and the shear angle γ_xy_: 0.3% for strain ε_x_, 0.37% for strain ε_y_, 0.22% for major strain ε_1_ and 0.06% for shear angle γ_xy_.

When analyzing the regression relations for the thickness reduction, it can be noticed that the coefficients of both the step down and the punch diameter in the mathematical expression of the strains are insignificant, the expression of the thickness reduction being only dependent on the wall angle. This does not come as a surprise, as it is known that in the case of metallic parts, there is a so-called “sine law”, whereby the value of the thickness reduction only depends on the wall angle of the part. Thus, it can be seen that even in polymer parts, the thickness reduction only depends on the wall angle.

### 3.5. Determination of the Main Strains and Thickness Reduction during Incremental Forming of PA and HDPE Sheets for Frustum of Pyramid-Shaped Parts

Two polyamide and polyethylene parts with a frustum of a pyramid-type trajectory (Figure 2b) were also made using the following input parameters: D_p_ = 8 mm, s = 0.75 mm and α = 50°, corresponding to Cases S2 and S11. Figure 15, Figure 16, Figure 17, Figure 18, Figure 19 and Figure 20 show the results obtained.

All the figures for the 18 cases and the video of the thickness reduction for the Cases S2 and S11 and for the pyramid frustum parts can be found as Appendix A.

Upon analyzing the images of the strains for the frustum of pyramid parts, it can be observed that the strains in the Ox and Oy directions have maximum values on two faces of the frustum of a pyramid, namely the faces directed towards the respective axes. The local maxima of these strains are found close to the edge of the frustum of a pyramid, located at the end of a linear trajectory. The maximum values of the major strain ε_1_ are concentrated near the four edges of the frustum of a pyramid. The maximum values of the minor strain ε_2_ are found, similar to the frustum of cone parts, concentrated towards the upper base of the frustum. The maximum and minimum values of the shear angle γ_xy_ are found on the faces of the frustum of a pyramid, with greater concentration towards the upper base of the frustum. As in the case of the frustum of cone-type parts, the thickness reduction of the frustum of pyramid-type parts has a variation similar to that of the major strain ε_1_.

The maximum values of the ε_x_ strain in the case of the frustum of pyramid-type parts are higher for polyamide (0.5439 mm/mm) than for polyethylene (0.456 mm/mm). The same conclusion is seen for the ε_y_ strain, where the strain for polyamide is 0.564 mm/mm and for polyethylene is 0.459 mm/mm. These values are higher than those obtained for the ε_x_ and ε_y_ strains in the frustum of cone-type parts: 0.5157 mm/mm (ε_x_) and 0.5150 mm/mm (ε_y_) for polyamide and 0.4190 mm/mm (ε_x_) and 0.4165 mm/mm (ε_y_) for polyethylene.

The maximum values obtained in the case of the frustum of pyramid-type parts for the major strain ε_1_ are also higher for both polyamide (0.5824 mm/m) and polyethylene (0.478 mm/mm) than those obtained for the frustum of cone-type parts (0.5215 mm/mm—polyamide; 0.448 mm/mm—polyethylene).

The maximum values obtained in the case of the frustum of pyramid-type parts for the minor strain ε_2_ are also higher for both polyamide (0.2022 mm/m) and polyethylene (0.2088 mm/mm) than those obtained for the frustum of cone-type parts (0.1409 mm/mm—polyamide; 0.1775 mm/mm—polyethylene).

Concerning the shear angle γ_xy_, the maximum values obtained for the frustum of pyramid-type parts are lower for both polyamide (17°) and polyethylene (14.7°) than those obtained for the frustum of cone-type parts (24.45°—polyamide; 17.5°—polyethylene).

The maximum values obtained in the case of the frustum of pyramid-type parts for the thickness reduction *t_r_* are higher for both polyamide (0.6465 mm/mm) and polyethylene (0.6060 mm/mm) than those obtained for the frustum of cone-type parts (0.5831 mm/mm—polyamide; 0.5827 mm/mm—polyethylene).

Since the processed parts were the frustums of cone- and pyramid-type parts, it should be noted that the strains for the two types of parts are different. Thus, if, for the frustum of a cone, the forming process is carried out under plane strain conditions, in the case of the frustum of a pyramid, it is carried out under plane strain conditions in the area of the walls and under biaxial strain conditions in the area of the corners of the part. Several authors [35] found that, as a result of the development of wrinkle defects, in polyamide, even in the case of the frustum of cone-type parts, the strain loading path on the wall area changes from plane strain to biaxial stretching conditions. This is not the case here, as the maximum angle of the wall is 60°, and thus no wrinkle defects occur.

In conclusion, it can be seen that, except for the shear angle, all the strains present in the case of the frustum of pyramid parts are larger than those of the frustum of cone parts. This is due to the geometry of the frustum of pyramid-type parts, with linear edges that form stress and strain concentrators. The shear angle has lower values in the case of the frustum of pyramid-type parts because the pyramidal shape gives better shear strength.

## 4. Conclusions

The main conclusions drawn from the paper, related to the influence of the wall angle, punch diameter and step down, are summarized in Table 18.

As far as the part material is concerned, all strains except for the minor strain and the thickness reduction are lower in the case of HDPE than in the case of PA. For the minor strain, the values are lower in the case of PA than in the case of HDPE, and for the thickness reduction, it is not possible to draw a conclusion related to the material because, as shown, it only depends on the wall angle.

As the aim of this paper was to find which optimal parameters lead to the decrease in the thickness reduction, it can be concluded that manufacturing with α = 50°, D_p_ = 10 mm and s = 0.75 mm results in a minimum thickness reduction for both types of materials.

## Figures and Tables

**Figure 1 materials-16-01644-f001:**
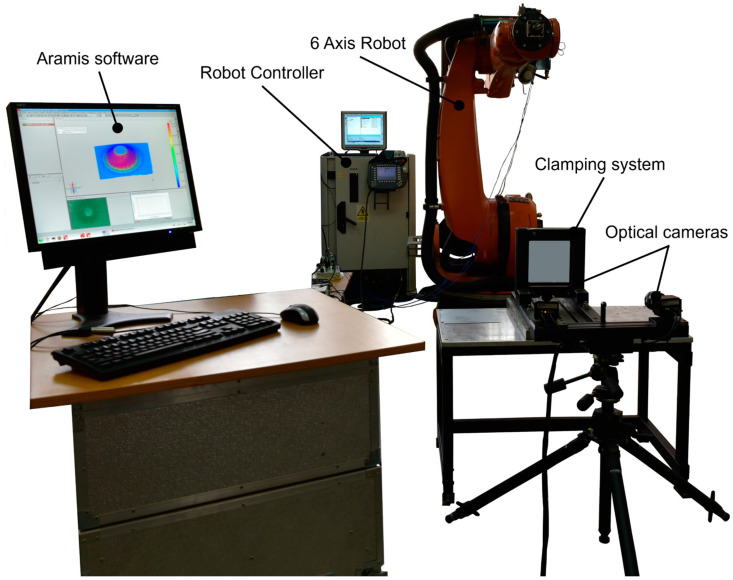
The experimental layout used for the measurement of the main strains.

**Figure 2 materials-16-01644-f002:**
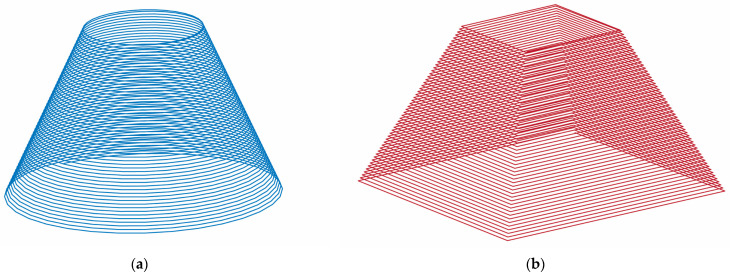
The two types of trajectories used in the experiments: (**a**)—conical trajectory; (**b**)—pyramidal trajectory.

**Figure 3 materials-16-01644-f003:**
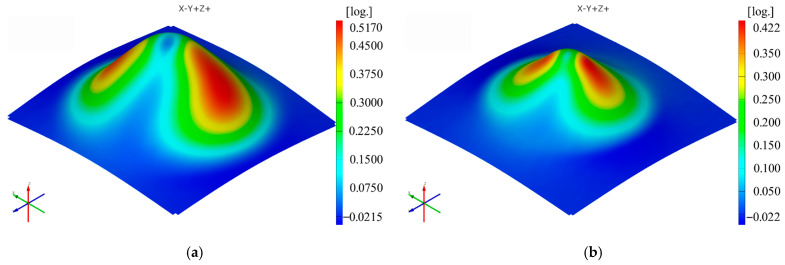
The ε_x_ strain distribution for the parts produced from PA (**a**) and HDPE (**b**), with Dp = 8 mm, s = 0.75 and α = 50° (Case S2).

**Figure 4 materials-16-01644-f004:**
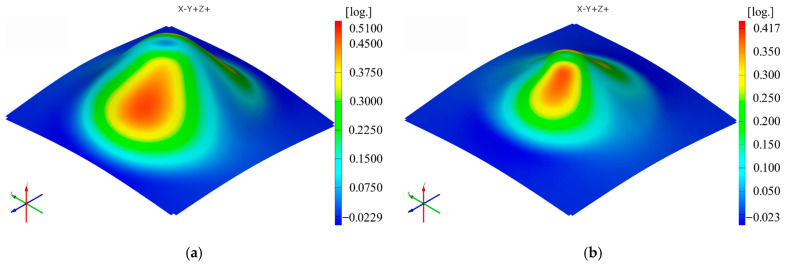
The ε_y_ strain distribution for the parts produced from PA (**a**) and HDPE (**b**), with Dp = 8 mm, s = 0.75 and α = 50° (Case S2).

**Figure 5 materials-16-01644-f005:**
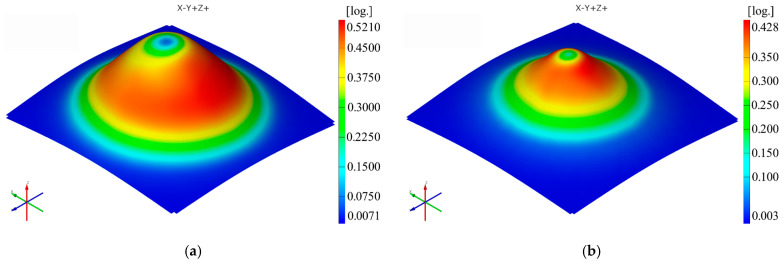
The major strain (ε_1_) distribution for the parts produced from PA (**a**) and HDPE (**b**), with Dp = 8 mm, s = 0.75 and α = 50° (Case S2).

**Figure 6 materials-16-01644-f006:**
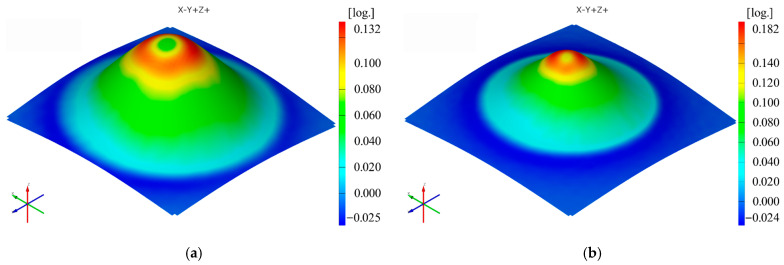
The minor strain (ε_2_) distribution for the parts produced from PA (**a**) and HDPE (**b**), with Dp = 8 mm, s = 0.75 and α = 50° (Case S2).

**Figure 7 materials-16-01644-f007:**
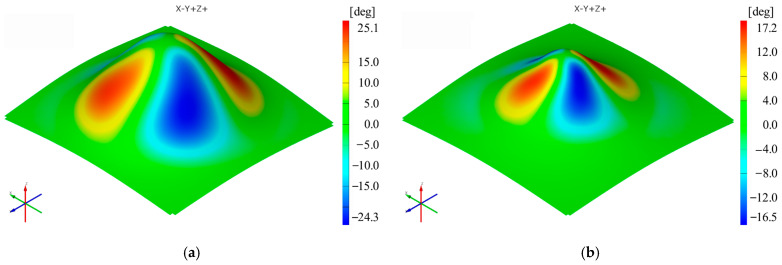
The shear angle (γ_xy_) distribution for the parts produced from PA (**a**) and HDPE (**b**), with Dp = 8 mm, s = 0.75 and α = 50° (Case S2).

**Figure 8 materials-16-01644-f008:**
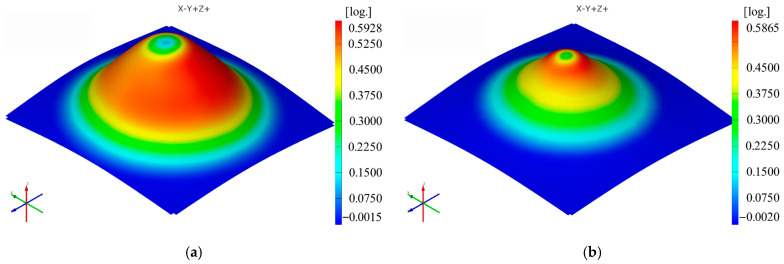
The thickness reduction (tr) distribution for the parts produced from PA (**a**) and HDPE (**b**), with Dp = 8 mm, s = 0.75 and α = 50° (Case S2).

**Figure 9 materials-16-01644-f009:**
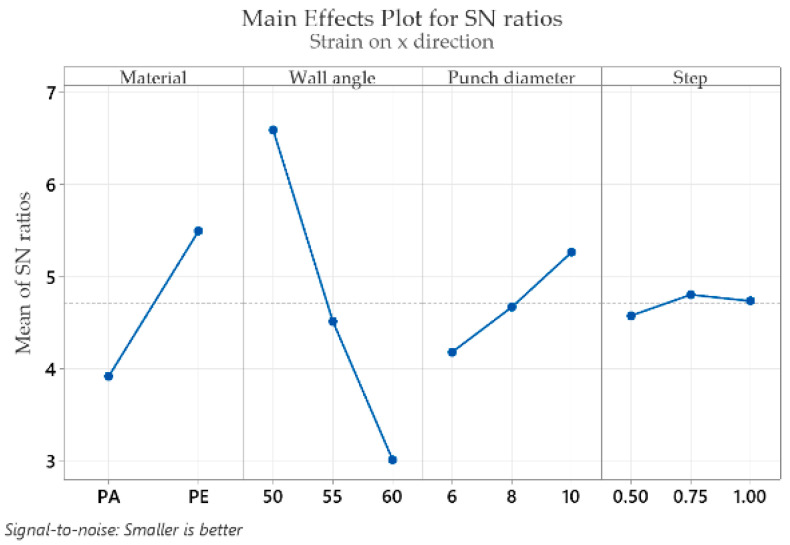
The main effects plot for S/N ratios for the strain in the x direction, ε_x_.

**Figure 10 materials-16-01644-f010:**
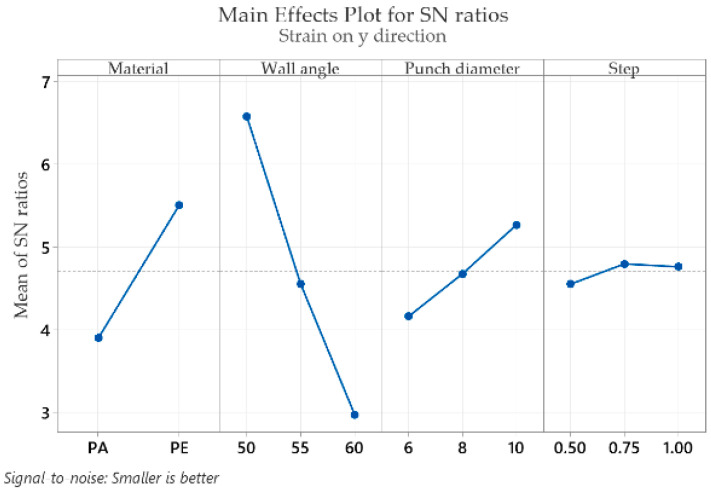
The main effects plot for S/N ratios for the strain in the y direction, ε_y_.

**Figure 11 materials-16-01644-f011:**
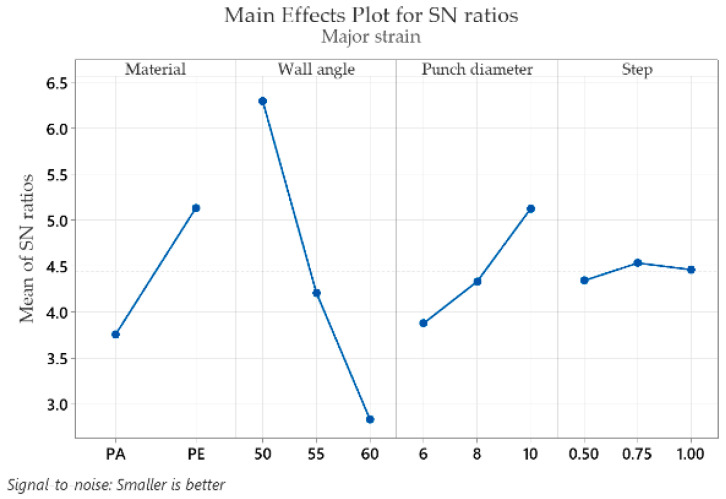
The main effects plot for S/N ratios for the major strain, ε_1_.

**Figure 12 materials-16-01644-f012:**
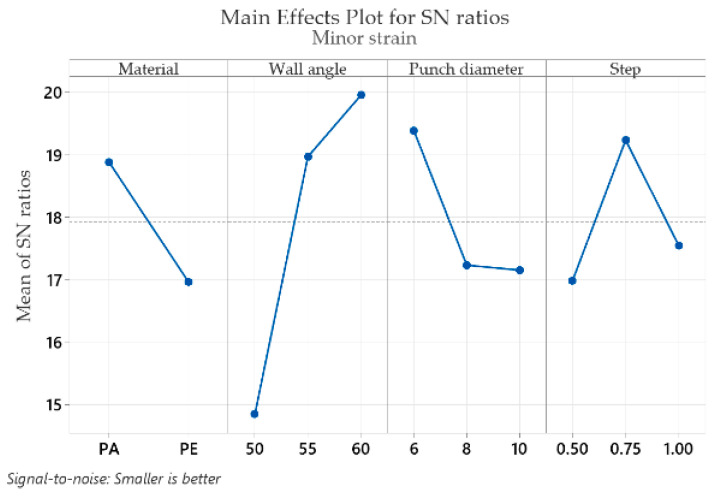
The main effects plot for S/N ratios for the minor strain, ε_2_.

**Figure 13 materials-16-01644-f013:**
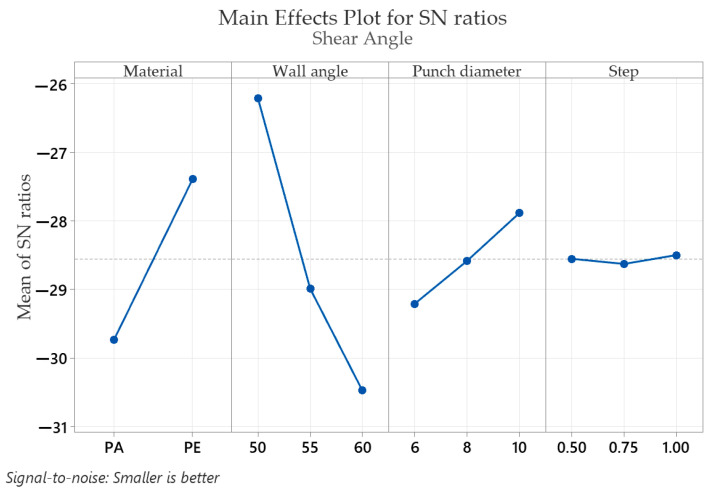
The main effects plot for S/N ratios for the shear angle, γ_xy_.

**Figure 14 materials-16-01644-f014:**
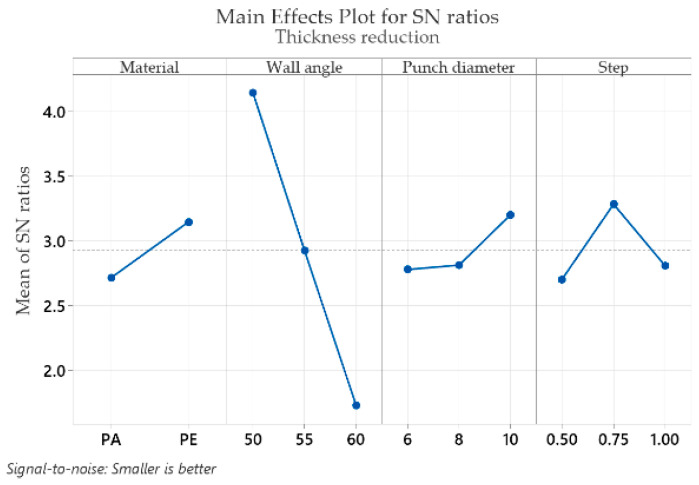
The main effects plot for S/N ratios for the thickness reduction, t_r_.

**Figure 15 materials-16-01644-f015:**
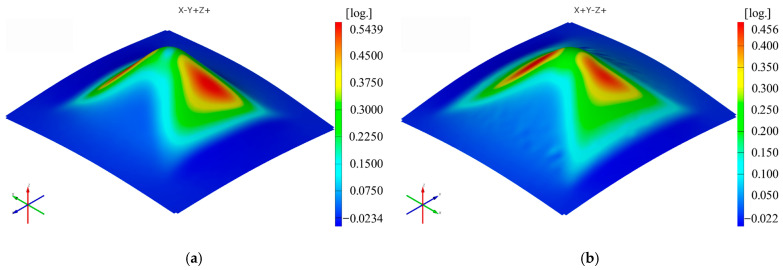
The ε_x_ strain distribution for the frustum of pyramid parts produced from PA (**a**) and HDPE (**b**), with Dp = 8 mm, s = 0.75 and α = 50°.

**Figure 16 materials-16-01644-f016:**
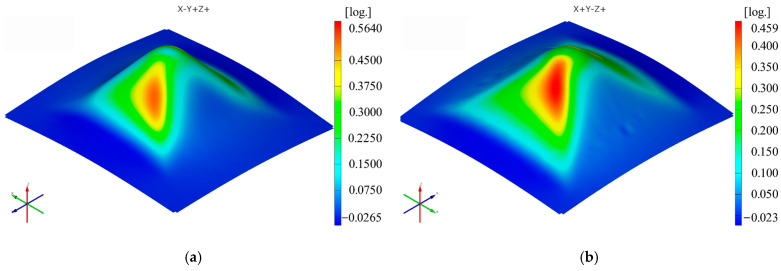
The ε_y_ strain distribution for the frustum of pyramid parts produced from PA (**a**) and HDPE (**b**), with Dp = 8 mm, s = 0.75 and α = 50°.

**Figure 17 materials-16-01644-f017:**
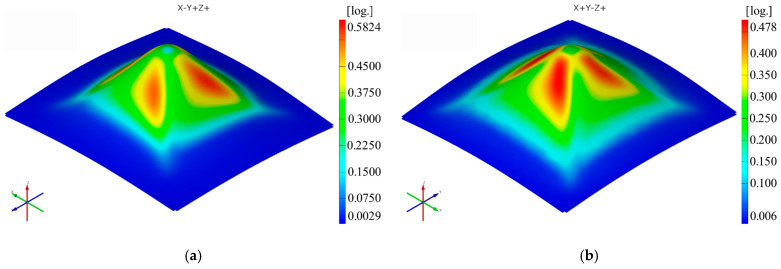
The major strain ε_1_ distribution for the frustum of pyramid parts produced from PA (**a**) and HDPE (**b**), with Dp = 8 mm, s = 0.75 and α = 50°.

**Figure 18 materials-16-01644-f018:**
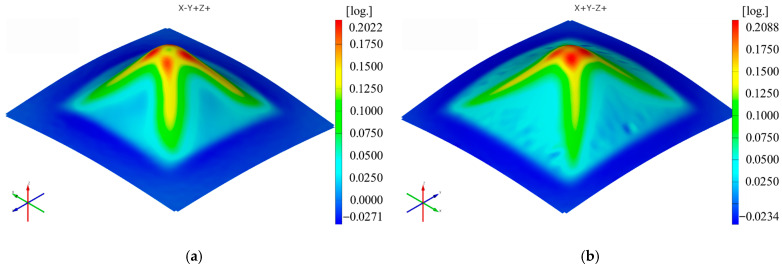
The minor strain ε_2_ distribution for the frustum of pyramid parts produced from PA (**a**) and HDPE (**b**), with Dp = 8 mm, s = 0.75 and α = 50°.

**Figure 19 materials-16-01644-f019:**
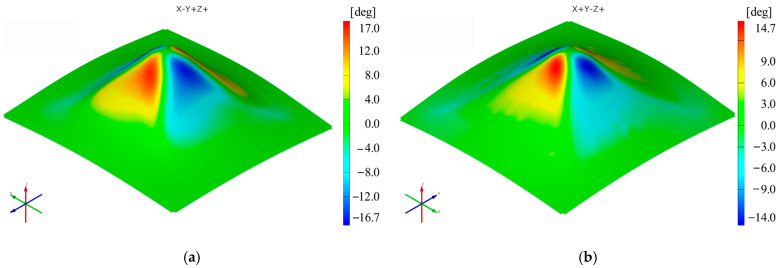
The shear angle γ_xy_ distribution for the frustum of pyramid parts produced from PA (**a**) and HDPE (**b**), with Dp = 8 mm, s = 0.75 and α = 50°.

**Figure 20 materials-16-01644-f020:**
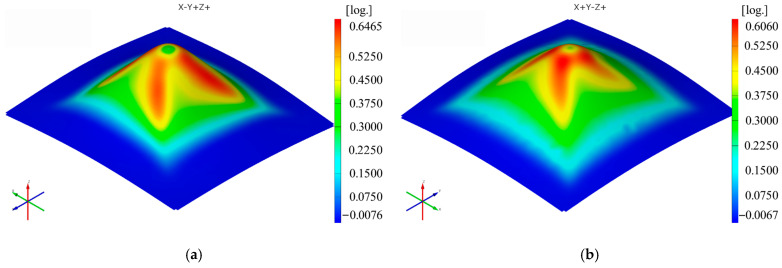
The thickness reduction t_r_ distribution for the frustum of pyramid parts produced from PA (**a**) and HDPE (**b**), with Dp = 8 mm, s = 0.75 and α = 50°.

**Table 1 materials-16-01644-t001:** The main influence factors and their level of variation.

Input Factors					Output Factors
Code	Type	Symbol	Unit	Variation Domain	y_1_, y_2_, y_3,_ y_4_, y_5,_ y_6_
*x_1_*	Polymeric material	*MAT*	-	PA, HDPE	ε_x_, ε_y_, ε_1_, ε_2,_ t_r_ [mm/mm] and γ_xy_ [°]
*x_2_*	Step down	*s*	(mm)	0.5, 0.75,1	ε_x_, ε_y_, ε_1_, ε_2,_ t_r_ [mm/mm] and γ_xy_ [°]
*x_3_*	Punch diameter	*D_p_*	(mm)	6, 8, 10	ε_x_, ε_y_, ε_1_, ε_2,_ t_r_ [mm/mm] and γ_xy_ [°]
*x_4_*	Wall angle	*α*	(°)	50, 55, 60	ε_x_, ε_y_, ε_1_, ε_2,_ t_r_ [mm/mm] and γ_xy_ [°]
*x_5_*	Initial thickness	*g*	(mm)	0.5, 2, 5	ε_x_, ε_y_, ε_1_, ε_2,_ t_r_ [mm/mm] and γ_xy_ [°]
*x_6_*	Feed rate	*v_s_*	(mm/min)	100, 1000, 2000	ε_x_, ε_y_, ε_1_, ε_2,_ t_r_ [mm/mm] and γ_xy_ [°]
*x_7_*	Punch angular speed	*ω*	(rot/min)	50, 120, 180	ε_x_, ε_y_, ε_1_, ε_2,_ t_r_ [mm/mm] and γ_xy_ [°]

**Table 2 materials-16-01644-t002:** The L18 (2^1^ × 3^3^) Taguchi orthogonal array used in experiments.

Case	Mat. Code	α Code	Dp Code	s Code	Mat.	α (°)	Dp (mm)	s (mm)
S1	−1	−1	−1	−1	PA	50	6	0.50
S2	−1	−1	0	0	PA	50	8	0.75
S3	−1	−1	1	1	PA	50	10	1.00
S4	−1	0	−1	0	PA	55	6	0.75
S5	−1	0	0	1	PA	55	8	1.00
S6	−1	0	1	−1	PA	55	10	0.50
S7	−1	1	−1	1	PA	60	6	1.00
S8	−1	1	0	−1	PA	60	8	0.50
S9	−1	1	1	0	PA	60	10	0.75
S10	1	−1	−1	−1	HDPE	50	6	0.50
S11	1	−1	0	0	HDPE	50	8	0.75
S12	1	−1	1	1	HDPE	50	10	1.00
S13	1	0	−1	0	HDPE	55	6	0.75
S14	1	0	0	1	HDPE	55	8	1.00
S15	1	0	1	−1	HDPE	55	10	0.50
S16	1	1	−1	1	HDPE	60	6	1.00
S17	1	1	0	−1	HDPE	60	8	0.50
S18	1	1	1	0	HDPE	60	10	0.75

**Table 3 materials-16-01644-t003:** The results and the statistical processing for the thickness reduction, t_r_.

Code	Mat.	α (°)	Dp (mm)	s (mm)	t_r,1_ (mm/mm)	t_r,2_ (mm/mm)	t_r_ Mean (mm/mm)	Standard Deviation	S/N Ratio	Coefficient of Variation
S1	PA	50	6	0.5000	0.6372	0.6287	0.6330	0.0060	3.9724	0.0095
S2	PA	50	8	0.7500	0.5928	0.5734	0.5831	0.0137	4.6839	0.0235
S3	PA	50	10	1.0000	0.6074	0.6454	0.6264	0.0269	4.0590	0.0429
S4	PA	55	6	0.7500	0.7380	0.7290	0.7335	0.0064	2.6918	0.0087
S5	PA	55	8	1.0000	0.7044	0.6954	0.6999	0.0064	3.0991	0.0091
S6	PA	55	10	0.5000	0.7473	0.7612	0.7543	0.0098	2.4493	0.0130
S7	PA	60	6	1.0000	0.8960	0.8632	0.8796	0.0232	1.1128	0.0264
S8	PA	60	8	0.5000	0.8860	0.9310	0.9085	0.0318	0.8308	0.0350
S9	PA	60	10	0.7500	0.8480	0.8270	0.8375	0.0148	1.5396	0.0177
S10	HDPE	50	6	0.5000	0.7047	0.6445	0.6746	0.0426	3.4104	0.0631
S11	HDPE	50	8	0.7500	0.5865	0.5788	0.5827	0.0054	4.6917	0.0093
S12	HDPE	50	10	1.0000	0.6375	0.6203	0.6289	0.0122	4.0276	0.0193
S13	HDPE	55	6	0.7500	0.6820	0.7224	0.7022	0.0286	3.0672	0.0407
S14	HDPE	55	8	1.0000	0.7740	0.7920	0.7830	0.0127	2.1242	0.0163
S15	HDPE	55	10	0.5000	0.6107	0.6357	0.6232	0.0177	4.1057	0.0284
S16	HDPE	60	6	1.0000	0.7479	0.7660	0.7570	0.0128	2.4180	0.0169
S17	HDPE	60	8	0.5000	0.8320	0.8610	0.8465	0.0205	1.4462	0.0242
S18	HDPE	60	10	0.7500	0.6614	0.7480	0.7047	0.0612	3.0235	0.0869

**Table 4 materials-16-01644-t004:** Response table for signal-to-noise ratios for the strain in the x direction ε_x_ (with the “smaller is better” condition).

Level	Material	Wall Angle	Punch Diameter	Step Down
1	3.915	6.587 *	4.180	4.575
2	5.492 *	4.512	4.668	4.803 *
3		3.011	5.263 *	4.733
Delta	1.577	3.576	1.083	0.227
Rank	**2**	**1**	**3**	**4**

* Optimal level.

**Table 5 materials-16-01644-t005:** Response table for signal-to-noise ratios for the strain in the y direction ε_y_ (with the “smaller is better” condition).

Level	Material	Wall Angle	Punch Diameter	Step Down
1	3.900	6.582 *	4.163	4.553
2	5.508 *	4.553	4.679	4.797 *
3		2.976	5.270 *	4.762
Delta	1.608	3.606	1.107	0.245
Rank	**2**	**1**	**3**	**4**

* Optimal level.

**Table 6 materials-16-01644-t006:** Response table for signal-to-noise ratios for the major strain ε_1_ (with the “smaller is better” condition).

Level	Material	Wall Angle	Punch Diameter	Step Down
1	3.759	6.293 *	3.877	4.343
2	5.132 *	4.211	4.332	4.533 *
3		2.833	5.127 *	4.461
Delta	1.373	3.460	1.249	0.190
Rank	**2**	**1**	**3**	**4**

* Optimal level.

**Table 7 materials-16-01644-t007:** Response table for signal-to-noise ratios for the minor strain ε_2_ (with the “smaller is better” condition).

Level	Material	Wall Angle	Punch Diameter	Step Down
1	18.88 *	14.85	19.38 *	16.98
2	16.96	18.96	17.23	19.23 *
3		19.95 *	17.15	17.55
Delta	1.92	5.10	2.23	2.25
Rank	**4**	**1**	**3**	**2**

* Optimal level.

**Table 8 materials-16-01644-t008:** Response table for signal-to-noise ratios for the shear angle γ_xy_ (with the “smaller is better” condition).

Level	Material	Wall Angle	Punch Diameter	Step Down
1	−29.73	−26.22 *	−29.21	−28.55
2	−27.39 *	−28.99	−28.59	−28.63
3		−30.47	−27.89 *	−28.50
Delta	2.34	4.25	1.32	0.13
Rank	**2**	**1**	**3**	**4**

* Optimal level.

**Table 9 materials-16-01644-t009:** Response table for signal-to-noise ratios for the thickness reduction *tr* (with the “smaller is better” condition).

Level	Material	Wall Angle	Punch Diameter	Step Down
1	2.715	4.141 *	2.779	2.702
2	3.146 *	2.923	2.813	3.283 *
3		1.729	3.201 *	2.807
Delta	0.431	2.412	0.422	0.580
Rank	**3**	**1**	**4**	**2**

* Optimal level.

**Table 10 materials-16-01644-t010:** Results of the analysis of variance for the strain in the x direction, ε_x_.

Source	Degree of Freedom	Adjusted Value of the Sum of Squares	F Value	*p*-Value	Contribution (%)
MAT	1	11.1881	80.88	0	20.47
α	2	38.6838	139.82	0	70.75
Dp	2	3.5296	12.76	0.007	6.46
s	2	0.1625	0.59	0.585	0.3
MAT×Dp	2	0.1426	0.52	0.622	0.26
MAT×s	2	0.1315	0.48	0.643	0.24
Residual error	6	0.83			1.52
Total	17	54.668			

**Table 11 materials-16-01644-t011:** Results of the analysis of variance for the strain in the y direction, ε_y_.

Source	Degree of Freedom	Adjusted Value of the Sum of Squares	F Value	*p*-Value	Contribution (%)
MAT	1	11.6346	60.38	0	20.69
α	2	39.2186	101.76	0	69.75
Dp	2	3.6843	9.56	0.014	6.55
s	2	0.2098	0.54	0.606	0.37
MAT×Dp	2	0.1799	0.47	0.648	0.32
MAT×s	2	0.1442	0.37	0.703	0.26
Residual error	6	1.1562			2.06
Total	17	56.2275			

**Table 12 materials-16-01644-t012:** Results of the analysis of variance for the major strain, ε_1_.

Source	Degree of Freedom	Adjusted Value of the Sum of Squares	F Value	*p*-Value	Contribution (%)
MAT	1	8.4813	78.96	0	16.51
α	2	36.4153	169.51	0	70.88
Dp	2	4.7971	22.33	0.002	9.34
s	2	0.111	0.52	0.621	0.22
MAT×Dp	2	0.5708	2.66	0.149	1.11
MAT×s	2	0.3557	1.66	0.268	0.69
Residual error	6	0.6445			1.25
Total	17	51.3755			

**Table 13 materials-16-01644-t013:** Results of the analysis of variance for the minor strain, ε_2_.

Source	Degree of Freedom	Adjusted Value of the Sum of Squares	F Value	*p*-Value	Contribution (%)
MAT	1	16.532	18.14	0.005	11.08
α	2	87.897	48.23	0	58.93
Dp	2	19.171	10.52	0.011	12.85
s	2	16.372	8.98	0.016	10.98
MAT×Dp	2	1.407	0.77	0.503	0.94
MAT×s	2	2.309	1.27	0.348	1.55
Residual error	6	5.468			3.67
Total	17	149.156			

**Table 14 materials-16-01644-t014:** Results of the analysis of variance for the shear angle, γ_xy_.

Source	Degree of Freedom	Adjusted Value of the Sum of Squares	F Value	*p*-Value	Contribution (%)
MAT	1	24.6132	1249.1	0	28.39
α	2	55.969	1420.2	0	64.55
Dp	2	5.2589	133.44	0	6.07
s	2	0.0504	1.28	0.345	0.06
MAT×Dp	2	0.536	13.6	0.056	0.62
MAT×s	2	0.1552	3.94	0.081	0.18
Residual error	6	0.1182			0.14
Total	17	86.7009			

**Table 15 materials-16-01644-t015:** Results of the analysis of variance for the thickness reduction, *t_r_*.

Source	Degree of Freedom	Adjusted Value of the Sum of Squares	F Value	*p*-Value	Contribution (%)
MAT	1	0.8345	2.49	0.166	3.41
α	2	17.4585	26.06	0.001	71.38
Dp	2	0.6598	0.98	0.427	2.7
s	2	1.1492	1.72	0.257	4.7
MAT×Dp	2	1.3406	2	0.216	5.48
MAT×s	2	1.0056	1.5	0.296	4.11
Residual error	6	2.0095			8.22
Total	17	24.4576			

**Table 16 materials-16-01644-t016:** The coefficients of the regression analysis of the thickness reduction for polyamide.

Variable	Coefficient	SE Coefficient	T Value	*p*-Value
Constant term	−0.633	0.149	−4.25	0.008
Punch diameter (Dp)	−0.00233	0.00614	−0.38	0.721
Step down (s)	−0.0599	0.0491	−1.22	0.277
Wall angle (α)	0.02610	0.00246	10.63	0.000

**Table 17 materials-16-01644-t017:** The coefficients of the regression analysis of the thickness reduction for polyethylene.

Variable	Coefficient	SE Coefficient	T Value	*p* Value
Constant term	0.032	0.328	0.10	0.925
Punch diameter (Dp)	−0.0147	0.0135	−1.09	0.326
Step down (s)	0.016	0.108	0.15	0.886
Wall angle (α)	0.01407	0.00541	2.60	0.048

**Table 18 materials-16-01644-t018:** Summarized influence of the wall angle, punch diameter and step down on the strains and thickness reduction.

	Wall Angle (α)	Punch Diameter (Dp)	Step Down (s)
Decrease in strains in x and y directions ε_x_ and ε_y_	As α decreases	As Dp increases	At the mean value of s
Decrease in major strain ε_1_	As α decreases	As Dp increases	At the mean value of s
Decrease in minor strain ε_2_	As α increases	As Dp decreases	At the mean value of s
Decrease in shear angle γ_xy_	As α increases	As Dp increases	As s increases
Decrease in thickness reduction t_r_	As α decreases	As Dp increases	At the mean value of s

## Data Availability

Not applicable.

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
