# Peer review of "Minimizing the Main Strains and Thickness Reduction in the Single Point Incremental Forming Process of Polyamide and High-Density Polyethylene Sheets"

_materials, 2023, doi:10.3390/ma16041644_

Round 1

Reviewer 1 Report

The authors in the present paper want to highlight the influence of each input parameter on the output parameters, but also to provide a guideline for minimizing the main strains of parts manufactured by SPIF made of polyamide and high-density polyethylene.

In the opinion of the reviewer, the paper presented is mostly well written and interesting, but it must overcome some important drawbacks. The authors should consider as a major changes: first, although the results are well presented there is no discussion of the results neither among the two materials presented (just a line, 685) nor with some results from the literature. Second, related to the previous one, the results are not supported by any physical explanation. The third one, despite all the statistical analysis carried out, most of the figures in those sections are not discussed, there are results without discussion, f.i. the Figure 18 which has a different trend compared to the rest presented in that section. Morover, to end from the statistical analysis, they do not provide a guideline as they were pointing out as an aim, but that they only provide a recommendation or conclusion (line 581). Finally, the conclusions section is just a summary of the main results that without a previous discussion of results are just that, a summary of the results.

Minor changes to consider:

-       The authors should use the typical nomenclature that will find for SPIF methodology: step down instead of step, feed rate instead of punch feed.

-       In Table 1, the reviewer recommends that the authors do not indicate the variation, they only put the three values that really consider, then you do not need to verify the value in the middle (0) in Table 2.

-       Maybe one Table from Tables 3-8 can be kept it in the main text, and the rest of the tables can be put at the end as Annex.

-       The reviewer recommends also that instead of putting Figures 3-14 grouped based on material (3-8 for PA, and 9-14 for HDPE) I would recommend presenting them grouped by conditions, in the way the odd figures will be for PA f. i. and even figure for HDPE. In that sense, the comparison could also be made easily by the reader and especially if it is also accompanied by a comparison paragraph in the text

Reviewer 2 Report

The manuscript “Minimizing the main strains and thickness reduction in the single point incremental forming process of polyamide and high-density polyethylene sheets “ is suitable for publication. The authors studied the influence of the most important parameters, namely the punch diameter, step on vertical direction, wall angle and material type on the most important strains that occur as a result of the SPIF process: strain on x and y direction, major strain, minor strain, shear angle and thickness reduction. The stain is hard to control in engineering practice and the job is interesting. I think this work expands the field of research. Stress-strain regulation is important to SPIF. 1.It would be better if the third part were more concise. 2.There are too many figures in the text . The figures of similar process conditions can be combined. For example, Figure.21 and Figure 22 are modified as “Figure 21. The strain distribution of εx (a) and εy (b) for the frustum of PA pyramid parts with Dp = 8 mm, s = 0.75 and α = 50°”.

Reviewer 3 Report

This work reported the analysis of the results obtained from the measurement of forces during the incremental forming of polyamide and high-density polyethylene sheets. The aim of this work was to minimize the main strains and thickness in the single point incremental forming process by mainly varying three parameters including punch diameter (6 mm, 8 mm and 10 mm), wall angle (50°, 55° and 60°) and step (0.5 mm, 0.75 mm and 1 mm). The experimental design is not complicated, but the description in the manuscript is too cumbersome, which makes it impossible to quickly understand the focus and significance of this research. Some further modifications and adjustments are needed, and some recommendations that should be reconsidered as follows:

1. Two materials (PA and HDPE) have been chosen for their low density, relatively good mechanical properties. The authors discussed the influence of equipment and process parameters such as wall angle and punch diameter on materials, but the different processability and mechanical properties of two kinds of materials have not been considered for discussion together.

2. The processing rheological properties, modulus and strength of different materials are closely related to the processing properties. The lack of laws in material structure, performance and processing also makes the scientific depth of this work insufficient, thus it looks more like an engineering experimental summary.

3. There are too many descriptions about experimental process in the manuscript. It is suggested to simplify them and highlight the key contents or results. Some data can be sorted into supplementary documents.

4. In the Conclusions part, it is suggested to organize the influence of various parameters on performance into a table form, which is more intuitive for comparative analysis.

5. There are many errors in English grammar or expression in the manuscript, which makes it less readable. It is suggested to revise the text carefully.
